# Innovations in Evaluating Statin Benefit and Efficacy in *Staphylococcus aureus* Intracellular Infection Management

**DOI:** 10.3390/ijms232113006

**Published:** 2022-10-27

**Authors:** Erik T. Nesson, Susan A. McDowell

**Affiliations:** 1Department of Economics, Ball State University, Muncie, IN 47306, USA; 2Office of Research and Innovation, Miami University, Oxford, OH 45056, USA

**Keywords:** intracellular infection, statins, *Staphylococcus aureus*

## Abstract

An emerging therapeutic approach in the treatment of infectious disease is to augment the host response through repurposing of well-tolerated, non-antibiotic, host-directed therapeutics. Earlier retrospective studies identify a positive association between statin use and a decreased risk of death due to sepsis or bacteremia. However, more recent randomized control trials fail to detect a therapeutic benefit in these complex infection settings. It is postulated that unrecognized biases in certain observational studies may have led to an overestimation of benefit and that statin use is instead a marker for health status, wealth, and demographic characteristics which may separately affect death due to infection. What remains unresolved is that in vitro and in vivo evidence reproducibly indicates that statin pharmacology limits infection and augments immunomodulatory responses, suggesting that therapeutic benefits may be attainable in certain infection settings, such as intracellular infection by *S. aureus*. Carefully considering the biological mechanisms capable of driving the relationship between statins and infections and constructing a methodology to avoid potential biases in observational studies would enable the examination of protective effects against infection and limit the risk of underestimating statin efficacy. Such an approach would rely on the examination of statin use in defined infection settings based on an underlying mode-of-action and pharmacology, where the inhibition of HMG-CoA-reductase at the rate-limiting step in cholesterol biosynthesis diminishes not only cholesterol levels but also isoprenoid intermediates central to host cell invasion by *S. aureus*. Therapeutic benefit in such settings, if existent, may be of clinical importance.

## 1. Introduction

An emerging approach in infectious disease management is the use of host-directed therapeutics that augment the host response to combat infection. For nearly two decades, researchers have examined statin drugs in this exploration of potential host-directed therapeutics for infection [1,2,3,4,5]. The underlying pharmacology of statins includes not only host responses to cholesterol lowering but also host responses to diminished intermediates within the cholesterol biosynthesis pathway that are exploited by invasive pathogens. Statin mode-of-action is through inhibition of 3-hydroxy-3-methyl-glutaryl-coenzyme A (HMG-CoA)-reductase at the early rate-limiting step within the cholesterol biosynthesis pathway [4] (Figure 1A). Inhibiting this early step diminishes cholesterol levels as well as intermediates within the cholesterol biosynthesis pathway. Intermediates such as farnesyl pyrophosphate and geranylgeranyl pyrophosphate operate in multiple systems. These long, hydrophobic chains are incorporated into small GTPases at the CaaX domain through post-translational prenylation, providing membrane anchors and sites for protein–protein interactions (Figure 1B). Statins limit formation of these intermediates, limiting post-translational prenylation of key regulatory molecules, including those exploited by pathogenic bacteria for host cell invasion. Prenylation-dependent and non-prenylation-dependent host responses are important determinants in immunomodulatory effects of statins as well (Table 1). Thus, underlying statin pharmacology is critical to consider in examining potential therapeutic benefits in infectious disease settings.

## 2. Supporting Evidence from In Vitro and In Vivo Studies

Early epidemiological studies that found that statins improved outcomes in sepsis and bacteremia launched an exploration into potential underlying mechanisms. In vitro and in vivo studies revealed reproducible mechanisms for efficacy in infection at physiologic concentrations, including immunomodulatory, anti-oxidative, and anti-inflammatory effects [4]. Statins eradicate latent intracellular infection by a range of pathogens through depleting cholesterol stores within infected host cells [1], forming phagocyte extracellular traps from host chromosomal DNA [21], and limiting host cell invasion through depletion of intermediates within the cholesterol biosynthesis pathway [11]. In vivo, statins decrease hematogenous spread and increase bacterial clearance [19,22,23]. Thus, statin pharmacology spans host-directed mechanisms with the potential to limit infection. 

## 3. Contradictory Evidence from Randomized Clinical Trial Data 

Ultimately, randomized clinical trials find that statins do not effectively treat sepsis or acute respiratory distress syndrome (ARDS) [24,25,26], though there are improved outcomes when there is prior exposure to statins [27]. More recent work suggests observational studies may overestimate the therapeutic benefit of statins due to biased selection criteria, bringing into question whether statins are efficacious in the treatment of infectious disease [28,29]. 

Observational studies may not be able to sufficiently control for potential confounding factors, and biases like healthy user bias or immortal time bias may account for the apparent protective effects of statins against infection in these studies. Immortal time bias may be introduced into observational studies examining the effects of medications when researchers define exposure to a drug as beginning with a prescription in a follow-up period. The period leading up to the prescription is called immortal time because anyone prescribed a drug at that time must be alive, thus leading to a bias in the groups classified as receiving a drug or not receiving a drug [30]. Healthy-user bias may arise if the population taking statins is healthier than the population not taking statins in uncontrolled-for dimensions. Previous research has suggested populations of patients taking statins differ from otherwise similar populations not taking statins. For example, Dormuth et al. [31] find that statin users in Canada are less likely to suffer workplace and motor vehicle accidents, even after controlling for other factors, suggesting unobservable factors that cause people to take statins are also related to health-promoting behaviors.

The underlying biological mechanisms by which statins protect against infection suggest statin use must commence before a possible infection or event that may cause a future infection. Initial exposure to statins after infection may not offer many benefits. Importantly, the randomized controlled trials which fail to find evidence for a protective effect of statins randomized patients into receiving statins after an infection diagnosis. Thus, it may be that randomized control trials were not able to create the setting in which statins are likely to confer protective effects against infection. In support of this notion is the report from Caffrey et al., who examine data from veterans and take care to account for biases, which may be present in previous observational work [3]. Limiting selection criteria to *S.*
*aureus* as the etiologic agent in bacteremia, a lower risk for bacteremia is associated with prevalent statin users, which are defined as people having statin therapy 30 days prior to the bacterial culture with no prior exposure in the preceding year who continued statin use; or for incident users, who did not continue statins, compared to non-statin users. These findings suggest a study design grounded on underlying statin pharmacology is crucial in determining whether statins are effective in reducing infections.

## 4. Settings Where Biology Suggests Statins May Offer Protection against Infection

### 4.1. Cardiac Device Infection

Between 1993 and 2008, cardiac implantable electronic device (CIED) procedures in the US doubled to nearly 4.2 million permanent pacemakers and implantable cardioverter-defibrillators [32]. Estimates of the risk of infection for primary implants is between 0.5–1.0%. For replacement CIED procedures, the risk of infection increases more than two-fold to 5% [33]. In a large cohort study of Medicare patients, CIED infection rates doubled the risk of death one-year post-implant [34]. Infection origination includes hematogenous spread and through development at the site of the implant, involving cardiac structures and evolution into infective endocarditis [33]. Onset can be acute, yet over 50% of infections are diagnosed at more than a year post-implantation. Staphylococcal strains predominate in acute and delayed infection onset. Increasing numbers of procedures and the likelihood of infection in longer-term use indicate an unmet medical need for improved infection management. 

Statin pharmacology may be especially effective in limiting CIED infection. Intravascular imaging revealed increased invasive capacity by *S. aureus* in vascular regions of high shear stress [35], such as those surrounding endovascular cardiac devices. In vivo evidence for intracellular infection was associated with elevated pathogenesis, indicating intracellular infection contributing to pathogenesis [36]. In these regions, circulating statins interact directly with endothelial cell layers, raising the possibility that statin inhibition of an invasion would limit the progression of endovascular infection. McAuley et al. maintain that patients with underlying endovascular disease may benefit from statin therapy in an infection setting [37]. However, epidemiological and medical research using observational studies on cardiac implant infection in patient populations on a statin regimen is minimal, but at least one study suggests that statin therapy offers protection against infection. Using a retrospective cohort analysis on a US veterans population, Alzahrani et al. [38] found that statin use at the time of initial CIED placement or revision is associated with a substantial reduction in CIED infections. 

### 4.2. Hip and Knee Replacement Infection

Hip and knee replacement surgery is an effective strategy to improve pain management and mobility for individuals suffering from a range of musculoskeletal issues [39]. In the US, the number of hip replacements more than doubled between 2000–2010 from an estimated 138,700 to an estimated 310,800 in patients aged 45 years and older [40]. Between 2000 and 2010, the number of total knee replacements nearly doubled to an estimated 693,400 in 2010, becoming the most frequent surgical procedure for patients aged 45 and older [41]. By 2030, the number of hip replacements is projected to increase to 572,000 and the number of knee replacements to 3.48 million [42]. The mean age of hip [40] and knee [41] replacement recipients has declined steadily, suggesting future increases will exceed current projections due to anticipated increases in the number of surgeries needed to replace aging prosthetics. 

Infection incidence has remained near 2% [43,44,45], indicating that increases in the number of surgeries will be accompanied by an increase in the number of infections. This apparent failure in infection control may be attributable to increases in at-risk populations, including patients with a diagnosis of diabetes [44]. Treatment requires lengthy rounds of antimicrobial therapy, progressing to surgical debridement, revision, and when incurable, amputation for survival [44]. Mortality rates from hip and knee replacement infections are estimated at 2–7% [44]. The risk for re-infection following revision is elevated, nearing 8% [46,47], and factors associated with hip revision include young age [48]. Advances are needed to address anticipated increases in the number of infected individuals and the ensuing personal and economic burden caused by these projected increases in the number of hip and knee replacement surgeries.

Infection characteristically is recalcitrant to antibiotic therapies regardless of whether the infecting strain is susceptible or resistant to the current standards of care [44]. Of the bacterial species isolated from infected hip and knee replacements, coagulase-negative staphylococci and *S. aureus* isolates predominate [49]. Infection due to methicillin-susceptible *S. aureus* (MSSA) is estimated to occur 2.5 times more frequently than infection due to MRSA, indicating that antibiotic resistance is not the sole driver for antibiotic treatment failure in the complex setting of joint infection. Etiology includes surgical site contamination and hematogenous spread due to transient bacteremia and sepsis [44]. It is postulated that intracellular bacterial reservoirs are an under-reported, less tractable infection source capable of remerging post-antimicrobial therapy to initiate infection at the site of an implanted device [6]. Prosthetic devices remain vulnerable to infection post-surgery, indicated by 50% of MSSA infection diagnoses occurring after treatment cessation at a median of 361 days post-implant surgery [49]. The propensity for antibiotic therapy to fail, the increases in the number of implant surgeries in younger populations, and the complex nature of prosthetic joint infection are indicative of an unmet need for new advances in therapeutic approaches to improve treatment outcomes.

Genotypic distinctions have been identified in infecting strains of *S. aureus*. Whereas fibronectin binding protein variants with increased invasive capacity have been linked to cardiac device infection [50,51], no such linkage was identified in prosthetic joint infection [52]. This distinction could have important ramifications for statin efficacy if limiting invasive capacity is a leading protective benefit of statin therapy. However, immunomodulatory effects may be of importance in either infection setting. 

## 5. Innovations to Evaluate Statin Efficacy 

As noted above, evidence regarding the relationship between statins and outcomes from bacteremia and sepsis are mixed between observational studies and randomized control trials [3,24,25,26]. Moreover, few papers examine whether statin use is associated with the risk of CIED and total knee and total hip replacement infection and revision. Lalmohamed et al. [53] and Cook et al. [54] examine the relationship between total hip replacement and total knee replacement revisions and statin use in the UK and Denmark, respectively. Alzahrani et al. [38] examine statins and CIED infections among US veterans. Here again, it is plausible that the effects found in this population do not translate to a broader population. 

Additionally, studies need to carefully consider biological mechanisms potentially affecting the relationship between statins and infection. In the studies mentioned above that examine the relationship between statin use and CIED and total knee replacement or total hip replacement infection and revision, statin use was measured at different points in time. Alzahrani et al. [38] focused on examining the potential protective effects of statin use on infection and measured statin use at the time leading up to the CIED procedure. However, statin use after the procedure was not examined. Lalmohamed et al. [53] and Cook et al. [54], on the other hand, do not consider statins as a protector of infection, but rather as a promoter of bone growth around an artificial knee or hip. Thus, Lalmohamed et al. [53] and Cook et al. [54] focused on statin use after the total knee replacement or total hip replacement procedure. Additionally, as mentioned above, biological mechanisms may play an important role in explaining why observational and randomized control trial results differ in studying whether statins offer protective effects in bacteremia and sepsis patients. Future studies are needed to carefully consider statin use during both the pre-operative period and post-operative period.

A related necessary innovation is carefully constructed methodologies to avoid potential biases in observational studies. Previous research into the relationship between statins and other medical conditions, for example, cancer, have been called into question for immortal time bias and healthy-user bias [29,55]. Thus, a study design that accounts for healthy-user bias is needed. Recent research offers strategies for ameliorating the above-mentioned biases. Some papers include controls for potential confounders [3]. Other papers go further by additionally creating a control group of non-statin users which is less likely to differ from the treatment group. In particular, Setoguchi et al. [55], who examine the relationship between statins and various forms of cancers, create a control group using Medicare beneficiaries on glaucoma medications. Farwell et al. [56] create a control group of people using antihypertensive medications but not cholesterol-lowering medications, which creates a larger control group. Some other papers use propensity score-matching methods, which match statin users to non-users in a way to minimize the potential of biases stemming from differences between the observable characteristics of the two groups [3,53]. Finally, recent research uses using time-dependent cohort designs to lessen the impact of immortal time bias [53].

A key consideration in the evaluation of efficacy will be whether the potential benefit in the infectious disease setting is adversely offset by the risk of development of adverse events, most notably, rhabdomyolysis. Muscle pain, weakness, and progression to associated morbidities and mortality through this muscle-weakening condition have been recognized as potential risk factors in statin use in infection settings [57,58,59]. Risk may be accentuated in severe infection and through drug interactions where underlying statin pharmacokinetics can lead to increased circulating statin concentrations [60]. In a delimited implanted medical device infection, this increased risk associated with altered statin pharmacokinetics may be diminished yet should remain as a risk factor, regularly evaluated in the examination of statin efficacy and safety. 

## 6. Summary Statement

In vitro and in vivo evidence indicates statin efficacy in the inhibition of intracellular infection and in augmentation of the host immune response. Studies controlling for known biases that examine statin use in biological settings where infection may be amenable to statin treatment may assist clinicians in decision-making regarding at-risk populations for acute and recurrent infection. The absence of such studies jeopardizes potential additional benefit in the use of well-tolerated statin therapies in the infection treatment arsenal.

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
