# Peer review of "Innovations in Evaluating Statin Benefit and Efficacy in Staphylococcus aureus Intracellular Infection Management"

_ijms, 2022, doi:10.3390/ijms232113006_

Round 1

Reviewer 1 Report

The title does not match the content of the manuscript. It must be changed.

The abstract should be rewritten

Line 7-11: the authors do not present the limitations of innovative therapeutic approaches in their manuscript but present data on statins mainly in Staphylococcus aureus infections.

Line 20-22: the data on the pharmacology and mode of action of statins in infections are not sufficiently detailed

Line 46: is it the legend of the diagram or the rest of the text; this is not clear

Diagram B concerns the potential action of statins on S aureus. It is simplistic and needs to be completed. Furthermore, it cannot be generalized to other pathogens.

Line 66-76: this paragraph is very inadequate. The data are incomplete. This important paragraph needs to be detailed and completely rewritten.

Line 78-106: the message of this paragraph is not clear. On this very interesting question of the potential benefit of statins in infections, the authors cannot limit themselves to a few references. They should either target their text on a pathogen, in particular Staphylococcus aureus which is most often cited in their review, or target a pathology

Line 116: the authors do not answer the question. The only data available are from the review 38.

Lines 128 to 138 explain the prevalence of S aureus infections in this context. This is therefore off-topic.

Line 143-163: these are epidemiological data but no data on the potential benefit of statins or other innovative therapies; this paragraph is too long for the title of the article.

Line 165-181: the authors only develop data on S aureus infections and the potential benefit of statins;

The data presented in this article are not in line with the titer. The authors present only  some data on the potential benefit of statins in particular models of S aureus infections. These data are incomplete

Reviewer 2 Report

In their review, the authors address therapeutic approaches to intracellular infection. In particular, the use of statins is discussed here. The review is very nicely written, I really enjoy reading it.

However, I strongly recommend adding a separate section, with its own figure, on rhabomyolysis.

Rhabdomyolysis is a clinically highly relevant side effect of statins, which also occurs particularly in the context of drug interactions and sepsis. This must be considered in any case during therapy, especially with regard to infections.

After an addition of the above facts, the article shows a comprehensive presentation on the clinically highly relevant topic. 
